# Targeted next-generation sequencing: a promising approach for *Mycobacterium tuberculosis* detection and drug resistance when applied in paucibacillary clinical samples

Wenting Jin,[1] Meixia Wang,[2] Yang Wang,[3] Beidi Zhu,[1] Qingqing Wang,[1] Chunmei Zhou,[4] Pei Li,[5] Chaohui Hu,[3] Jun Liu,[3] Jue Pan,[1] Jiachang Chen,[3] Bijie Hu[1]

**ABSTRACT**  Tuberculosis (TB) returns to be the leading infectious killer globally after coronavirus disease 2019. The World Health Organization (WHO) formally included targeted next-generation sequencing (tNGS) in its list of recommendations for *Mycobacterium tuberculosis* (MTB) and drug resistance (DR). In this study, we explored the application of various clinical sample types for TB diagnosis and DR profiles. In comparison to the composite reference standard, the overall sensitivity values of culture, Xpert, metagenomic next-generation sequencing (mNGS), and tNGS were 0.458, 0.614, 0.772, and 0.760, respectively. tNGS had sensitivity similar to mNGS, which had advantages over culture and Xpert, respectively, despite different classification of sample types. In comparison to the microbiological reference standard, the overall sensitivity values of culture, Xpert, mNGS, and tNGS were 0.606, 0.811, 0.856, and 0.884, respectively. Suprisingly, in extrapulmonary tissue and serous effusion, mNGS and tNGS had advantages over Xpert. DR-related mutations were detected in 15 cases (13.2%). There were 51 (44.7%) applicable for all DR genes, with 22 (19.3%) not applicable for DR genes. DR genes were partially applicable in 41 (36.0%) samples. However, in culture-negative TB cases, tNGS can additionally provide 52.7% first-line DR profiles. Sanger sequencing was performed on 14 samples to confirm gene mutation identified by tNGS, and the results were entirely consistent. It was concluded that the tNGS assay was a promising approach in the initial diagnostic test of MTB and DR-related genes in different clinical samples, even for the smear- and culture-negative paucibacillary samples.

**IMPORTANCE** tNGS combines gene-specific amplification with next-generation sequencing to detect MTB and drug-resistant genes by amplifying numerous loci directly from clinical samples. The WHO implemented tNGS for the purpose of monitoring respiratory specimens for MTB detection and DR-TB due to its high sensitivity and specificity, culture independence, and ability to report heterogeneous/silent mutations. The sensitivity outperformed both culture and Xpert, and the turnaround time was significantly less than that of culture-based assays. The tNGS assay used in this study costs USD 96 and has a 12 hour turnaround time. Nonetheless, tNGS has a great deal of promise for enhancing TB detection while also addressing DR strains.

**KEYWORDS**  tuberculosis, *Mycobacterium tuberculosis*, targeted next-generation sequencing, paucibacillary tuberculosis

Tuberculosis (TB) returns to be the leading infectious disease killer globally after coronavirus disease 2019, surpassing HIV (1). Globally, there were an expected 10.8 million new cases of tuberculosis in 2023, a 2.2% rise over 2020 (1). In China, despite significant advancements in the eradication of TB, the incidence was 52 per 100,000

**Peer Reviewers** Fangyou Yu, Tongji University Affiliated Shanghai Pulmonary Hospital, Shanghai, China; Bo Yan, Shanghai Public Health Clinical Center, Shanghai, China

Address correspondence to Jiachang Chen, zb-chenjiachang@kingcreate.com.cn, or Bijie Hu, hu.bijie@zs-hospital.sh.cn.

Wenting Jin, Meixia Wang, and Yang Wang contributed equally to this article. The authorship order was determined by relative intellectual contribution across hypothesis formulation, methodology development, and data interpretation, as documented in the contributorship statement.

Jiachang Chen and Bijie Hu contributed equally to this article.

The authors declare no conflict of interest.

See the funding table on p. 11.

in 2023. The low etiological confirmation and the high drug-resistant burden remain to be very difficult problems. Effective management and prevention of TB depend on the timely and accurate diagnosis of *Mycobacterium tuberculosis* complex (MTBC) and its drug susceptibility. As the primary healthcare setting for most TB patients seeking initial care (2), general hospitals face significant challenges in diagnosing both pulmonary tuberculosis (PTB) and extrapulmonary tuberculosis (EPTB).

Culture-based methods remain the gold standard for *Mycobacterium tuberculosis* (MTB) diagnosis and phenotypic drug susceptibility testing (pDST)(3). The slow bacterial growth rate, complex operational procedures, and stringent-level biosafety requirements make culture-based detection methods incapable of meeting the need for swift clinical detection. Therefore, it is critical to accelerate the development and application of diagnostic methods for TB, especially those that can directly identify pathogens in clinical samples. Molecular approaches have reduced TB diagnosis times to a few days, even to a few hours. The World Health Organization (WHO) recommended a number of nucleic acid detection techniques, including Xpert MTB/rifampicin (RIF) (4) (referred to as Xpert; Cepheid, USA) and loop-mediated isothermal amplification, due to their high sensitivity and specificity for MTB diagnosis, while it was also widely acknowledged that these techniques were obviously ineffective for detecting paucibacillary specimens.

As for molecular drug susceptibility testing (mDST), whole-genome sequencing (WGS) of clinical MTB isolates allows for more accurate identification of all chromosomal mutations. WGS has great performance for profiling resistance of first- and second-line drugs (5, 6), which is widely adopted. Nevertheless, WGS is only applicable for MTB isolates but not clinical samples. Xpert MTB/RIF Ultra (Cepheid) was recommended by the WHO in 2017, presenting a substantially lower limit of detection (LoD) than Xpert due to the use of multi-copy *IS1081* and *IS6110* insertion elements as MTB target sequences, and both of them have similar performance in detecting rifampicin resistance. Xpert MTB/XDR (Cepheid) can detect mutations associated with resistance to isoniazid (INH), ethambutol (EMB), fluoroquinolone (FQN), and second-line injectable drugs (7, 8). However, Xpert MTB/XDR's target genes are limited, and it is unable to report mutation types. Usually, it is necessary to first test Xpert or Xpert Ultra as a supplement to more drug resistance (DR) information in MTB confirmed or rifampicin-resistant confirmed samples.

Targeted next-generation sequencing (tNGS) combines gene-specific amplification with next-generation sequencing (NGS) to detect MTB and drug-resistant genes by amplifying numerous loci directly from clinical samples. According to the results of a systematic review and meta-analysis (9), tNGS has a 94.1% overall sensitivity and a 98.1% specificity for all drugs. For first-line drugs and some second-line drugs, such as fluoroquinolones(FQN) and amikacin (AMK), tNGS has especially high accuracy. The WHO has published an implementation guideline (10) for the use of tNGS in monitoring DR-TB because it offers several benefits, such as being culture independent, being directly relevant to clinical samples, and capable of reporting heterogeneous/silent mutations (11, 12). It might be a powerful culture-free alternative to routine laboratory / traditional tests. While previous studies were mainly focused on application in clinical isolates and smear positive/respiratory clinical samples (13–16), the diagnosis of TB in paucibacillary specimens is actually more nettlesome. Our previous work concluded that metagenomic next-generation sequencing (mNGS) (17) may be a promising technology for the early auxiliary diagnosis of MTB, especially in sputum-negative pulmonary TB and tuberculous serous effusion. However, it cannot detect DR genes and is costly.

In this study, we explored the application of various clinical sample types for TB diagnosis, focusing particularly on the detection of MTB and its DR genes across a range of clinical samples. We compared the efficacy of MTB identification with culture, Xpert, and mNGS. Furthermore, we evaluated tNGS's performance in the detection of DR mutations.

## MATERIALS AND METHODS

### Study design and participants

This study was carried out at Zhongshan Hospital, Fudan University, which is a comprehensive, tertiary-level hospital with 3,000 beds. Figure 1 displays the study design flowchart. A total of 227 susceptible TB participants were selected from June 2021 to August 2024, of which 178 participants met the inclusion criteria. The Xpert MTB/RIF (Cepheid), liquid culture (MGIT 960, BD, USA), acid-fast bacilli (AFB), and histopathology were all parts of standard diagnostic processes. Clinical samples were stored at −80°C and were tested for mNGS and MTB-tNGS (referred to as tNGS). Microbiological reference standard (MRS) referred to positive culture or a positive nucleic acid test for MTBC (including Xpert and TB-PCR). Composite reference standard (CRS) referred to a patient who presented with symptoms, signs, images, microbial results, or pathological findings suggestive of TB, where a clinician has diagnosed TB and decided to treat the patient

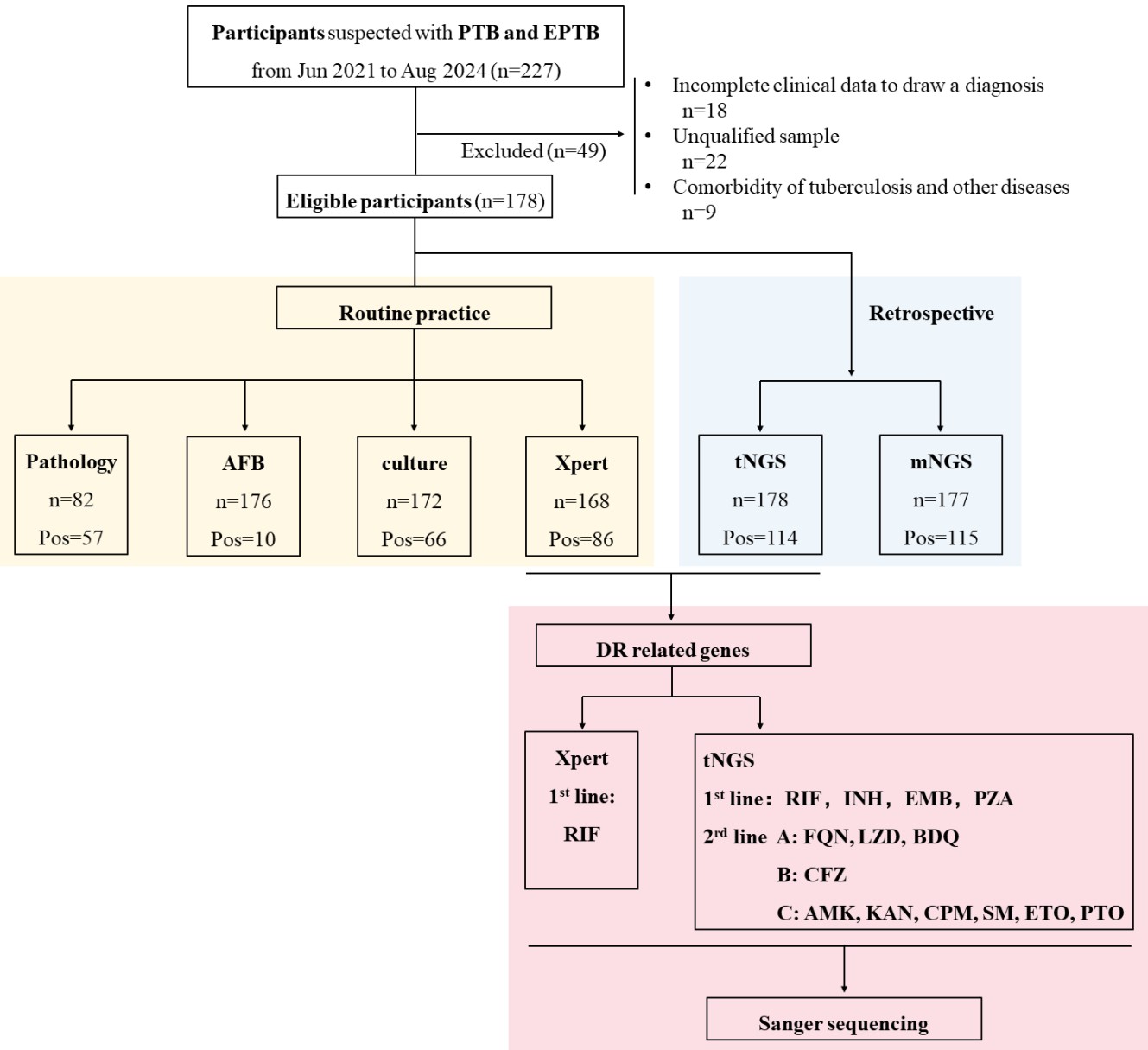

**FIG 1** Flowchart of the study design.

with a full course of TB therapy. The CRS standard complied with the People's Republic of China Health Industry Standard (WS 288–2017).

## Xpert MTB/RIF assay

An equal volume of sample processing reagent was combined with the homogenate of lymph node tissue, vortexed for 15 seconds, and then left to stand at room temperature for 15 minutes. The GeneXpert Infinity System (Cepheid) was then loaded with a sample of 2 mL of the processed liquid that was added to the Xpert reaction reagent cartridge (Xpert, Cepheid). The findings were automatically read out by the system after 2 hours.

## MTB liquid culture

The samples were to be digested, decontaminated, and concentrated using the Clinical and Laboratory Standards Institute standard methodology. After processing, 0.5 mL of the material was utilized to quickly cultivate mycobacteria (MGIT 960, BD). On a positive culture sample, MPB64 antigen detection and acid-fast staining confirmation were carried out.

## mNGS

The mNGS process was performed on a rapid on-site platform in the hospital, as described previously (18). In short, clinical samples (0.5 mL) were treated to produce DNA fragments prior to the extraction of genomic DNA. After that, DNA libraries were created using the BGISEQ-2000 platform, and a bioinformatics pipeline was used to examine the sequencing results. The RefSeq database was used to align the resulting data. According to the preceding description, the RefSeq was deemed positive if at least one read was mapped to the MTBC (number of sequences rigorously aligned at the genus level ≥1).

## tNGS

Prior to nucleic acid extraction, sputum samples and viscous BALF were treated to achieve liquefaction. Fresh tissue samples were minced and homogenized by oscillation. Processed samples were aliquoted (1.3 mL) and subjected to high-speed centrifugation. The supernatant was then removed, retaining approximately 500 µL of the sample. A 490 µL portion was combined with 10 µL of an exogenous internal control and processed in a tissue homogenizer for mechanical disruption. The mixture was subsequently centrifuged at 12,000 rpm for 5 minutes, and 250 µL of the supernatant was collected for nucleic acid extraction using appropriate extraction or purification reagents (Guangzhou KingCreate Biotechnology Co. Ltd., Guangzhou, China).

Library construction was performed using an MTBC and drug-resistance gene Extraction Kit (KingCreate Co. Ltd.). The extracted nucleic acids were enriched for target regions and underwent library purification steps to complete the library construction process. Nuclease-free water was used as a non-template control to monitor for contamination.

Generated libraries were quantified using the Equalbit DNA HS Assay Kit (Vazyme Biotech, Nanjing, Jiangsu, China) with Invitrogen Qubit v.3.0/4.0 (Thermo Fisher Scientific, Waltham, MA, USA) Fluorometer to ensure all samples were with a library density of ≥0.5 ng/µL or else the library should be subjected to re-construction. The constructed libraries were pooled to a homogeneous mass. The size of the library fragments was determined by an automated nucleic acid protein analyzer (Qsep100) using Standard Cartridge Kit (S2). The size of the library fragments should be from 250 to 350 bp. The qualified pooled library was diluted and denatured, 500 µL of which was subjected to KM MiniSeq Dx-CN Platform (KingCreate Co. Ltd.) for sequencing.

Generated sequencing raw read data underwent quality control procedures. The fastp v.0.20.1 (3) was employed for adapter trimming (19) and low-quality filtering using default parameters followed by reference-based assembly using bwa v.0.7.17 in "mem" mode (20). The tNGS assay interrogates 18 genes using 631 amplicons from loci

associated with MTBC and resistance to 13 anti-TB drugs: RIF (*rpoB*), INH (*inhA* and *katG*), pyrazinamide (*pncA*), EMB (*embB* and *embA*), fluoroquinolones (*gyrA* and *gyrB*), linezolid (*rplC* and *rrl*), bedaquiline (*atpE* and *Rv0678*), clofazimine (*Rv0678*), SM (*rrs*, *rpsL*, and *qid*), KAN and AMK (*rrs* and *eis*), CPM (*rrs* and *tlyA*), ETO (*ethA* and *inhA*), and PTO (*inhA*), mainly from Genbank and RefSeq. To call positive signals for specific pathogens, mapped reads were counted and normalized to reads per 100,000 mapped reads (RPhK). Cases with specific RPhKs were considered as positive for each sample. If a specific species or a higher-level taxonomy unit was identified in a sample with an RPhK value of ≥3, the species/unit was regarded as "present" in this sample or else reported as "absent."

## Sanger sequencing

The extracted nucleic acid was added to the Sanger reaction mixture for PCR amplification. After amplification, the product was purified and electrophoresed to confirm the target band, which was then excised and used for Sanger sequencing to obtain sequence data.

## Statistical analysis

Continuous variables were described as mean ± standard deviation and median and interquartile range according to data distribution. The Mann-Whitney *U* test or *t*-test was used to compare the characteristics between two groups. The diagnostic effectiveness was assessed by computing the sensitivity, specificity, accuracy, and area under the curve value for various methods. Kappa analysis was used to describe the level of data consistency. A *P* value of <0.05 (two-tailed) was considered to be statistically significant. Statistical analyses were performed using the R v.3.6.1 software.

## RESULT

### Characteristics of participants and samples

A total of 178 participants (105 male/73 female, average age of 55.1 ± 17.84, with a male-to-female ratio of 1.44:1.0) were included in this study. The flowchart of the study is shown in Fig. 1. Among all the samples, 10 were AFB positive. Due to the diverse sources of the samples, we classified them in three ways: respiratory/non-respiratory, tissue/non-tissue, as well as more detailed sputum, BALF, lung tissue, extrapulmonary tissue, serosal fluid, and pus. The specific distribution is shown in Tables 1 and 2 and Table S1 and S2.

### Performance of different methods in detecting of MTB

The overall sensitivity values of culture, Xpert, mNGS, and tNGS were 0.458 (95% confidence interval [CI] 0.377–0.540), 0.614 (95% CI 0.534–0.695), 0.772 (95% CI 0.704–0.839), and 0.760 (95% CI 0.692–0.828), respectively, in comparison to CRS. They all had a specificity value of 1. tNGS had similar sensitivity as mNGS, which had advantages over culture and Xpert, respectively, despite different classifications of sample types. Outstandingly, the advantage was more pronounced and tNGS even surpassed mNGS in serous fluid, as shown in Table 1 and Table S1. The overall sensitivity values of culture, Xpert, mNGS, and tNGS were 0.606 (95% CI 0.514–0.697), 0.811 (95% CI 0.737–0.886), 0.856 (95% CI 0.791–0.921), and 0.884 (95% CI 0.825–0.943), respectively, in comparison to MRS, as shown in Table 2 and Table S2. Culture and Xpert both had a specificity value of 1, while mNGS and tNGS had specificity values of 0.697 (95% CI 0.586–0.808) and 0.773 (95% CI 0.672–0.874). In the sputum, BALF, and pulmonary tissue samples, the sensitivity value of Xpert was not significantly lower than those of tNGS and mNGS while maintaining a high specificity value of 1. However, in extrapulmonary tissue and serous effusion, mNGS and tNGS had advantages over Xpert, although the had relatively low specificity.

**TABLE 1** The diagnostic performance of four methods for tuberculosis in diverse sample types in comparison to CRS standard[a,b]

| CRS as standard | Method | Sensitivity (95% CI) | Specificity (95% CI) | Accuracy (95% CI) | AUC (95% CI) | P value |
|---|---|---|---|---|---|---|
| All (n = 178) | Culture | 0.458 (0.377–0.540) | 1.000 | 0.547 (0.544–0.549) | 0.729 (0.688–0.770) | 0.000 |
| | Xpert | 0.614 (0.534–0.695) | 1.000 | 0.679 (0.676–0.681) | 0.807 (0.767–0.847) | 0.000 |
| | mNGS | 0.772 (0.704–0.839) | 1.000 | 0.808 (0.806–0.810) | 0.886 (0.852–0.920) | 0.000 |
| | tNGS | 0.760 (0.692–0.828) | 1.000 | 0.798 (0.796–0.800) | 0.880 (0.846–0.914) | 0.000 |
| Sputum (n = 33) | Culture | 0.731 (0.560–0.901) | 1.000 | 0.788 (0.778–0.798) | 0.865 (0.780–0.951) | 0.000 |
| | Xpert | 0.864 (0.720–1.000) | 1.000 | 0.897 (0.890–0.903) | 0.932 (0.860–1.000) | 0.000 |
| | mNGS | 0.920 (0.814–1.000) | 1.000 | 0.938 (0.934–0.941) | 0.960 (0.907–1.000) | 0.000 |
| | tNGS | 0.962 (0.888–1.000) | 1.000 | 0.970 (0.968–0.971) | 0.981 (0.944–1.000) | 0.000 |
| BALF (n = 17) | Culture | 0.462 (0.191–0.733) | 1.000 | 0.588 (0.560–0.616) | 0.731 (0.595–0.866) | 0.956 |
| | Xpert | 0.667 (0.400–0.933) | 1.000 | 0.750 (0.727–0.773) | 0.833 (0.700–0.967) | 0.015 |
| | mNGS | 0.846 (0.650–1.000) | 1.000 | 0.882 (0.870–0.894) | 0.923 (0.825–1.000) | 0.002 |
| | tNGS | 0.769 (0.540–0.998) | 1.000 | 0.824 (0.807–0.840) | 0.885 (0.770–0.999) | 0.005 |
| Pulmonary tissue (n = 28) | Culture | 0.391 (0.192–0.591) | 1.000 | 0.481 (0.463–0.500) | 0.696 (0.596–0.795) | 0.939 |
| | Xpert | 0.739 (0.560–0.919) | 1.000 | 0.778 (0.765–0.790) | 0.870 (0.780–0.959) | 0.003 |
| | mNGS | 0.792 (0.629–0.954) | 1.000 | 0.821 (0.811–0.832) | 0.896 (0.815–0.977) | 0.001 |
| | tNGS | 0.583 (0.386–0.781) | 1.000 | 0.643 (0.627–0.659) | 0.792 (0.693–0.890) | 0.019 |
| Extrapulmonary tissue (n = 55) | Culture | 0.500 (0.352–0.648) | 1.000 | 0.569 (0.559–0.578) | 0.750 (0.676–0.824) | 0.007 |
| | Xpert | 0.568 (0.422–0.715) | 1.000 | 0.627 (0.618–0.636) | 0.784 (0.711–0.857) | 0.003 |
| | mNGS | 0.771 (0.652–0.890) | 1.000 | 0.800 (0.794–0.806) | 0.885 (0.826–0.945) | 0.000 |
| | tNGS | 0.771 (0.652–0.890) | 1.000 | 0.800 (0.794–0.806) | 0.885 (0.826–0.945) | 0.000 |
| Serous effusion (n = 34) | Culture | 0.259 (0.094–0.425) | 1.000 | 0.394 (0.380–0.408) | 0.630 (0.547–0.712) | 0.922 |
| | Xpert | 0.286 (0.118–0.453) | 1.000 | 0.412 (0.398–0.426) | 0.643 (0.559–0.727) | 0.934 |
| | mNGS | 0.571 (0.388–0.755) | 1.000 | 0.647 (0.634–0.660) | 0.786 (0.694–0.877) | 0.007 |
| | tNGS | 0.679 (0.506–0.852) | 1.000 | 0.735 (0.724–0.747) | 0.839 (0.753–0.926) | 0.002 |

[a]Note: P < 0.05 in AUC compared to CRS standard. Eleven pus samples were not listed separately in the table because of the small sample size.
[b]AUC, area under the curve; BALF, bronchoalveolar lavage fluid; CI, confidence interval.

## Overlap of positive results from different methods and performance of combined methods

We compared the four methods with pathology in tissue samples, revealing that only 10 positive (13.5%) results were observed across all five methods, with a Fleiss kappa value of 0.329 (P < 0.001), as depicted in Fig. S1. Furthermore, we tried to analyze the combined performance of Xpert and tNGS with reduced turnaround time and improved sensitivity and specificity. Whether utilizing CRS or MRS as standards, the combined method performed nearly flawlessly in sputum samples, outperforming other sample types, as indicated in Tables S3 and S4.

## Comparison of MTB sequencing numbers in tNGS and mNGS based on different Xpert semi-quantitative groups

Eighty samples had consistent MTB identification findings between Xpert and tNGS, while 77 samples had consistent MTB identification results between Xpert and mNGS. We attempted to draw additional conclusions about the range of MTB sequencing numbers in various Xpert semi-quantitative groups (Fig. 2). In all samples and respiratory samples, there were notable differences in tNGS sequencing levels between the Xpert very low group and the other groups, including Xpert low and Xpert high groups. It is interesting to note that in all samples and respiratory samples, the only significant changes in mNGS sequencing levels were observed between the Xpert high group and others. However, in the non-respiratory samples, there was no significant difference in sequencing levels of tNGS or mNGS across different Xpert groups.

TABLE 2 The diagnostic performance of four methods for tuberculosis in diverse sample types in comparison to MRS[a,b]

| MRS as standard | Method | Sensitivity (95% CI) | Specificity (95% CI) | Accuracy (95% CI) | AUC (95% CI) | P value |
|---|---|---|---|---|---|---|
| All (n = 178) | Culture | 0.606 (0.514–0.697) | 1.000 | 0.750 (0.748–0.752) | 0.803 (0.757–0.849) | 0.000 |
| | Xpert | 0.811 (0.737–0.886) | 1.000 | 0.881 (0.880–0.882) | 0.906 (0.868–0.943) | 0.000 |
| | mNGS | 0.856 (0.791–0.921) | 0.697 (0.586–0.808) | 0.797 (0.795–0.798) | 0.776 (0.688–0.865) | 0.000 |
| | tNGS | 0.884 (0.825–0.943) | 0.773 (0.672–0.874) | 0.843 (0.841–0.844) | 0.828 (0.748–0.909) | 0.000 |
| Sputum (n = 33) | Culture | 0.792 (0.629–0.954) | 1.000 | 0.848 (0.841–0.856) | 0.896 (0.815–0.977) | 0.000 |
| | Xpert | 0.905 (0.779–1.000) | 1.000 | 0.931 (0.927–0.935) | 0.952 (0.890–1.000) | 0.000 |
| | mNGS | 0.957 (0.873–1.000) | 0.889 (0.684–1.000) | 0.938 (0.934–0.941) | 0.923 (0.778–1.000) | 0.000 |
| | tNGS | 1.000 | 0.889 (0.684–1.000) | 0.970 (0.968–0.971) | 0.944 (0.842–1.000) | 0.000 |
| BALF (n = 17) | Culture | 0.667 (0.359–0.975) | 1.000 | 0.824 (0.807–0.840) | 0.833 (0.679–0.987) | 0.003 |
| | Xpert | 0.889 (0.684–1.000) | 1.000 | 0.938 (0.930–0.945) | 0.944 (0.842–1.000) | 0.000 |
| | mNGS | 0.889 (0.684–1.000) | 0.625 (0.290–0.960) | 0.765 (0.744–0.785) | 0.757 (0.487–1.000) | 0.986 |
| | tNGS | 0.889 (0.684–1.000) | 0.750 (0.450–1.000) | 0.824 (0.807–0.840) | 0.819 (0.567–1.000) | 0.006 |
| Pulmonary tissue (n = 28) | Culture | 0.450 (0.232–0.668) | 1.000 | 0.593 (0.575–0.601) | 0.725 (0.616–0.834) | 0.018 |
| | Xpert | 0.850 (0.694–1.000) | 1.000 | 0.889 (0.882–0.896) | 0.925 (0.847–1.000) | 0.000 |
| | mNGS | 0.800 (0.625–0.975) | 0.625 (0.290–0.960) | 0.750 (0.737–0.763) | 0.713 (0.457–0.968) | 0.985 |
| | tNGS | 0.700 (0.499–0.901) | 1.000 | 0.786 (0.774–0.797) | 0.850 (0.750–0.950) | 0.001 |
| Extrapulmonary tissue (n = 55) | Culture | 0.647 (0.486–0.808) | 1.000 | 0.765 (0.758–0.772) | 0.824 (0.743–0.904) | 0.000 |
| | Xpert | 0.735 (0.587–0.884) | 1.000 | 0.824 (0.818–0.829) | 0.868 (0.793–0.942) | 0.000 |
| | mNGS | 0.838 (0.719–0.957) | 0.667 (0.449–0.884) | 0.782 (0.776–0.788) | 0.752 (0.584–0.921) | 0.000 |
| | tNGS | 0.892 (0.792–0.992) | 0.778 (0.586–0.970) | 0.855 (0.850–0.859) | 0.835 (0.689–0.981) | 0.000 |
| Serous effusion (n = 34) | Culture | 0.538 (0.267–0.809) | 1.000 | 0.818 (0.809–0.827) | 0.769 (0.634–0.905) | 0.000 |
| | Xpert | 0.615 (0.351–0.880) | 1.000 | 0.853 (0.846–0.86) | 0.808 (0.675–0.940) | 0.000 |
| | mNGS | 0.769 (0.540–0.998) | 0.714 (0.521–0.908) | 0.735 (0.724–0.747) | 0.742 (0.531–0.953) | 0.004 |
| | tNGS | 0.846 (0.650–1.000) | 0.619 (0.411–0.827) | 0.706 (0.694–0.718) | 0.733 (0.531–0.935) | 0.005 |

[a]Note: P < 0.05 in AUC compared to MRS. Eleven pus samples were not listed separately in the table because of the small sample size.
[b]AUC, area under the curve; BALF, bronchoalveolar lavage fluid; CI, confidence interval.

## Performance of tNGS and Xpert in detecting DR-related genes

There were 114 (64.0%) positive tNGS results out of 178 samples in detecting MTB. Ninety-nine of these satisfied the MRS, and all of them satisfied the CRS standard. Additionally, there were 51 (44.7%) applicable of all DR genes in our test, with 22 (19.3%) not applicable for all drugs. DR genes were partially applicable in 41 (36.0%) of the samples. Of these, 24 were first-line drug applicable; 9 were first-line plus second-line group A applicable; and 8 were first-line plus second-line groups A and B applicable, as shown in Fig. 3A. While in culture-negative TB cases, tNGS can additionally provide 52.7% first-line DR information. In Xpert-negative TB cases, tNGS can additionally provide 44.7% first-line drug resistance information, as shown in Fig. 3B.

DR-related mutations were detected in 15 cases (13.2%) with single or poly-DR, as shown in Fig. 4. Rifampicin resistance was detected in four cases, while three of them were "MTB detected and rifampicin resistance not detected" using Xpert. The tNGS results for PT80 revealed a rifampicin resistance mutation site at *rpoB* Leu553Pro with a mutation frequency of 100%. INH, FQN, SM, ETO, and PTO resistance mutation sites were also detected by tNGS in other samples. Except for the PT128 sample with QLN *gyrA* with a mutation frequency of 71%, all the other mutation sites have a mutation frequency of 100%. Sanger sequencing was performed on 14 samples to confirm gene mutation identified by tNGS. The results were entirely consistent with tNGS detection. The sample of PT80 could not be confirmed because of insufficient specimens.

## DISCUSSION

A timely diagnosis is essential to managing TB effectively. Despite significant progress in the identification and treatment of TB, a significant number of cases—including DR-TB—remain undiagnosed and untreated (21). Conventional detection techniques, however,

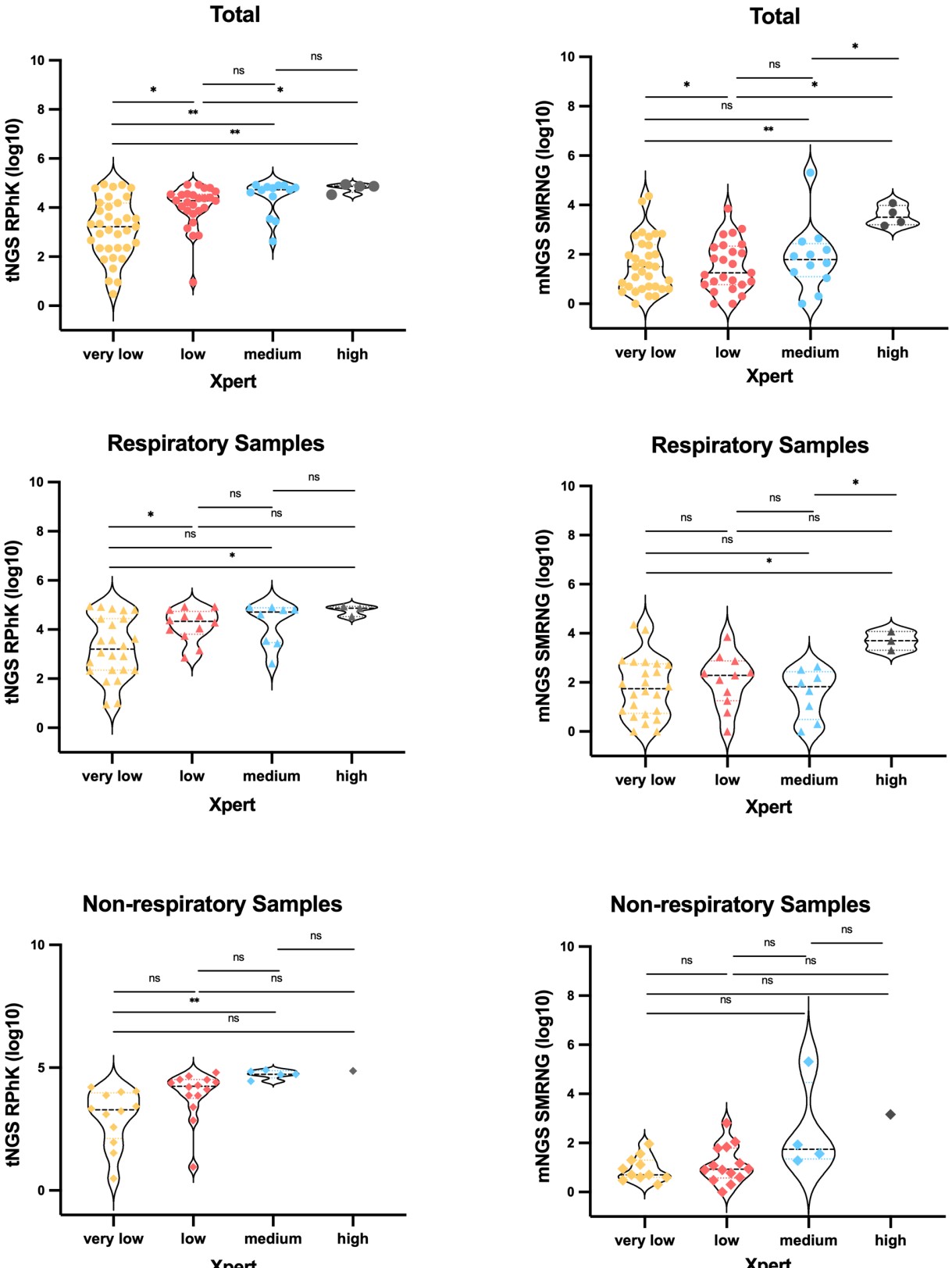

**FIG 2** Scatter plot of TB sequencing (log 10) detected by tNGS and mNGS based on different Xpert semi-quantitative categories. mNGS, metageonomic next-generation sequencing; NGS, targeted next-generation sequencing; ns, not significant; *, $P < 0.05$；**, $P < 0.01$; RPhK, reads per 100,000 mapped reads; SMRNG, number of sequences rigorously aligned at the genus level; Xpert, Xpert MTB/RIF.

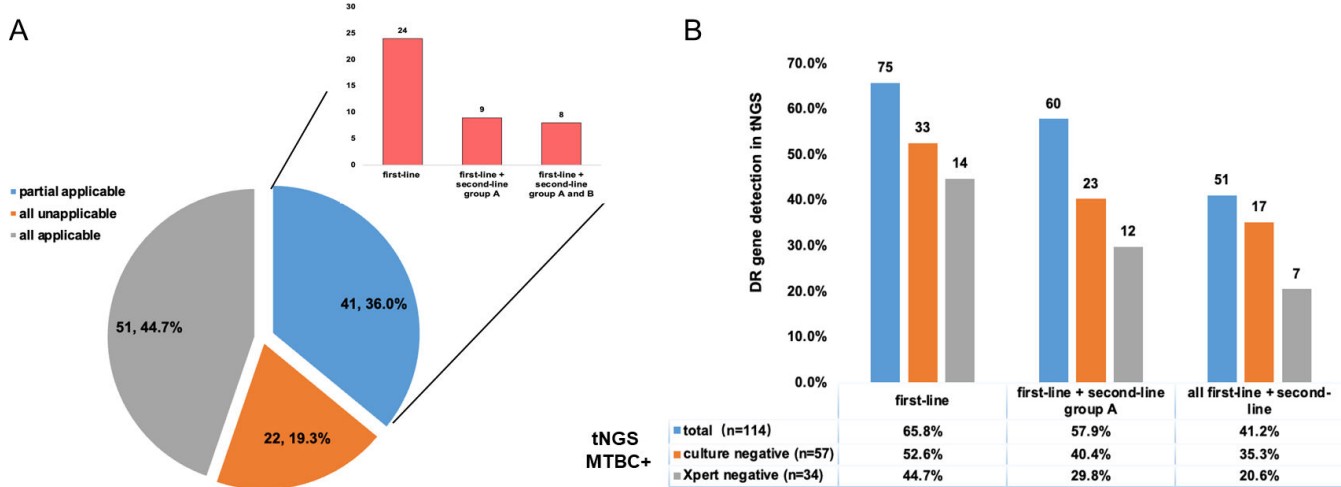

**FIG 3** Drug-resistant-related mutations detected in 114 tuberculosis samples (A and B). MTBC, *Mycobacterium tuberculosis* complex.

are deemed inadequate to meet clinical requirements because of misdiagnosis and delayed diagnosis (22). Several molecular techniques, like Xpert, despite their exceptional significance, perform poorly in diagnosing MTB when evaluating paucibacillary samples (23).

NGS technology has advanced rapidly in recent years and has been applied to identify pathogens and guide-targeted antimicrobial therapy (24–26). Our previous research confirmed that mNGS has an advantage in detecting MTB compared to culture (17). Similar results were also obtained in this study. With minor variations between sample types, the average sensitivity of tNGS in identifying MTB in this study is comparable to that of mNGS while surpassing that of Xpert and liquid culture. Yin et al. (27) found that multiplex PCR-based tNGS was not superior to mNGS. This could be because the investigation focused on more than 1,400 pathogen targets, including DNA and RNA, and tNGS was not TB specific. Another reason may be fewer TB cases in their study. Our results were comparable to those of BALF or sputum in previous studies where most of the samples tested positive for AFB. Diagnosing tuberculosis in patients with negative AFB or paucibacillary samples is more challenging, though. Applications in less frequent sample types, such as gastric aspirate (28), feces (29), cerebrospinal fluid (30), and formalin-fixed and paraffin-embedded tissue (31), have also been the subject of certain investigations. The use of tNGS is still a good source of DR profiles. The benefit of tNGS over Xpert and even mNGS in EPTB samples (particularly serous fluids) with fewer MTB was more substantial. On the contrary, the sensitivity of mNGS and Xpert exceeded that of tNGS in pulmonary tissue and BALF. Perhaps sensitivity may be significantly impacted by one to two positive results as a result of relatively modest BALF and lung tissue sample size. The fact that this tNGS pipeline was developed and optimized using sputum samples, attaining 100% sensitivity in sputum results, could be another factor. The pre-processing procedure for various specimen types may, however, differ slightly, necessitating additional modifications to enhance the performance of other specimen types. The bacteriological confirmation rate for TB patients worldwide in 2023 was 62% (1). With 55% in 2020 and 58% in 2021, it was still marginally below the global average in China (32). The WHO still believes that the use of rapid diagnostic tests remains far too limited (1). According to the most recent WHO guidelines, tNGS assay performance may be less effective for smear-negative samples, indicating that tNGS of smear-positive samples may be a more practical and economical method. Our findings demonstrate that tNGS exhibits exceptional sensitivity and specificity in detecting MTB from diverse clinical samples, including sputum, BALF, serious effusions, tissues, and pus. These results support the potential of tNGS for the initial diagnostic test of MTB detection in both PTB and EPTB.

| | Xpet MTB | Xpet RIF-R | RIF | INH | EMB | PZA | FQN | SM | ETO | PTO | tNGS RPhK | Sanger |
|---|---|---|---|---|---|---|---|---|---|---|---|---|
| PT7 | | | | | | | gyrA: Asp94Asn | | | | 19358 | Asp94Asn |
| PT36 | | rpoB + | rpoB: Asp516Val | katG: Ser315Thr | | | | rpsL: Lys43Arg | | | 70863 | Asp516Val Ser315Thr Lys43Arg |
| PT41 | | | | inhA: -15C>T; Ser94Ala | | | | rpsL: Lys43Arg | inhA: -15C>T; Ser94Ala | inhA: -15C>T; Ser94Ala | 63304 | -15C>T Ser94Ala Lys43Arg |
| PT57 | | | | katG: Ser315Asn | | | | | | | 28211 | Ser315Asn |
| PT58 | | | | inhA: -15C>T | | | gyrA: Ala90Val | rpsL: Lys88Arg | inhA: -15C>T | inhA: -15C>T | 73387 | -15C>T Ala90Val Lys88Arg |
| PT64 | | | | katG: Ser315Thr | | | | | | | 27068 | Ser315Thr |
| PT75 | | | | | | | | rpsL: Lys43Arg | | | 11131 | Lys43Arg |
| PT85 | | | | | | | gyrA: Ala90Val Asp94Asn | | | | 51858 | Ala90Val Asp94Asn |
| PT97 | | rpoB + | rpoB: Leu533Pro | | | | | rpsL: Lys43Arg | | | 84750 | Leu533Pro Lys43Arg |
| PT123 | | | | | | | | rpsL: Lys43Arg | | | 62554 | Lys43Arg |
| PT128 | | | | | | | gyrA: Asp94Gly | | | | 87653 | Asp94Gly |
| PT138 | | rpoB + | rpoB: Ser531Leu | | | | | rpsL: Lys43Arg | | | 32512 | Ser531Leu Lys43Arg |
| PT28 | | | | inhA: -15C>T | | | | rpsL: Lys43Arg | inhA: -15C>T | inhA: -15C>T | 2283 | -15C>T Lys43Arg |
| PT80 | | rpoB - | rpoB: Leu533Pro | | | | | | | | 867 | N/A |
| PT90 | | | | katG: Ser315Thr | | | | | | | 858 | Ser315Thr |

**Xpert MTB**
- N/A
- Negative
- Extremely low
- Low
- Medium
- High

**tNGS DR mutation**
- N/A
- DR-mutation not detected
- DR-mutation detected

**tNGS DR mutation**
- some detected
- all detected

**FIG 4** Drug-resistant-related mutations in 15 cases using Xpert MTB/RIF, tNGS, and Sanger sequencing. EMB, ethambutol; ETO, ethionamide; FQN, fluoroquinolone; INH, isoniazid; PTO, prothionamide; PZA, pyrazinamide; RIF, rifampicin; RPhK, mapped reads per 100,000.

It is somewhat challenging to assess paucibacillary samples for the identification of DR profiles. The gold standard for determining drug sensitivity in TB is still pDST (3). WGS can detect information at the sequence level, such as insertions, deletions, or rare mutations, which makes it more sensitive and specific in predicting DR-TB (33). Unfortunately, the requirement for an initial positive TB culture to produce a substantial bacterial load currently limits the effective application of pDST and WGS in clinical settings (34). On the other hand, tNGS has become a viable option for thorough, quick, and clinically relevant sequencing from many sample types in source-limited situations. It can be carried out on direct clinical samples. Many previous studies have confirmed the performance of tNGS in detecting DR-TB, whether in clinical isolates or clinical samples (13, 28, 31, 35–39). tNGS has a generally satisfactory sensitivity and specificity for all drugs, particularly first-line drugs, according to a systematic review and meta-analysis (9). A recent multi-center investigation (40) showed that the sensitivity of detecting RIF, quinolone, and pyrazinamide resistance was greatly decreased with low or very low detection in the Xpert group. Also, the study mentioned that in 102 AFB negative samples, tNGS still produced either partial or full drug resistance profiles. While the author did not go into greater detail, AFB alone is not sufficiently predictive to support downstream tNGS decisions. Our data were quite similar. In non-TB designated institutions or in EPTB cases, most will encounter these challenges (41, 42). Many cases were diagnosed with TB during their first visit to the hospital without the ability to identify DR characteristics. It could only be empirically treated as sensitive TB, which could result in drug resistance or treatment failure. How may DR profiles be obtained from paucibacillary clinical samples, particularly in cases that are culture and Xpert negative and lack a foundation for comparing pDST or *rpoB*? To what extent does tNGS detect the DR genes accurately? In our study, DR mutations were found in 14 of the total samples, including those with heteroresistance and those where resistance mutations were identified at low sequencing depths. These findings were fully consistent with those from Sanger sequencing, confirming that tNGS maintains accuracy at its unique

ability to detect heterogeneous mutations. Unfortunately, we were unable to confirm the sample with low tNGS RPhK, which discovered the *rpoB* gene because there were not enough samples. Zhang et al. (13) used an RIF-R clinical isolate to show that tNGS had a marginally lower LoD in culture medium than Xpert. Sanger sequencing verified that a mutation at the Ser315Thr of the *KatG* gene was present in another tissue sample with an RPhK of 858. Thus, our results support tNGS as a reliable method for identifying DR-TB, especially for those with a full DR file applicable.

Our study is still subject to some limitations. First, due to the retrospective nature of this investigation, bias in patient selection is unavoidable. Second, the low detection rate of DR-TB in this study led to limitations in the accuracy of resistance prediction for this tNGS assay. Third, the validity of our resistance data from tNGS may be compromised by the lack of WGS or pDST.

Overall, tNGS holds strong potential for improving TB diagnosis, simultaneously for DR strains. Its high efficiency, accuracy, and comprehensive detection capabilities provide valuable support for the precise control and treatment of TB, contributing to the ultimate goal of ending the TB epidemic.

## AUTHOR AFFILIATIONS

[1]Department of Infectious Diseases, Zhongshan Hospital, Fudan University, Shanghai, China
[2]Department of Infection Management, Zhongshan Hospital (Xiamen), Fudan University, Xiamen, Fujian, China
[3]Guangzhou KingCreate Biotechnology Co., Ltd., Guangzhou, Guangdong, China
[4]Department of Microbiology, Zhongshan Hospital, Fudan University, Shanghai, China
[5]KingMed Diagnostics, Guangzhou, Guangdong, China

## AUTHOR ORCIDs

Wenting Jin http://orcid.org/0000-0003-3333-8721
Qingqing Wang http://orcid.org/0009-0009-8846-0071
Jiachang Chen http://orcid.org/0009-0001-5331-6442
Bijie Hu http://orcid.org/0000-0002-8687-992X

## FUNDING

| Funder | Grant(s) | Author(s) |
| --- | --- | --- |
| Development Fund of Zhongshan Hospital, Fudan University | 2024ZSFZ39 | Bijie Hu |
| Development Fund of Zhongshan Hospital, Fudan University | XK-079-4 | Wenting Jin |

## AUTHOR CONTRIBUTIONS

Wenting Jin, Conceptualization, Data curation, Funding acquisition, Supervision, Writing – original draft | Meixia Wang, Data curation, Formal analysis, Methodology, Project administration, Writing – original draft | Yang Wang, Investigation, Methodology, Project administration, Writing – original draft | Beidi Zhu, Data curation, Investigation, Methodology, Project administration, Writing – original draft | Qingqing Wang, Data curation, Investigation, Methodology, Project administration | Chunmei Zhou, Data curation, Investigation, Methodology, Resources | Chaohui Hu, Methodology, Project administration | Jun Liu, Methodology, Project administration | Jue Pan, Conceptualization, Methodology, Project administration, Writing – review and editing | Jiachang Chen, Methodology, Supervision, Visualization, Writing – review and editing | Bijie Hu, Conceptualization, Methodology, Supervision, Visualization, Writing – review and editing.

## DATA AVAILABILITY

The targeted next-generation sequencing data have been made available for download on the NCBI SRA under study accession number PRJNA1238033, which is referenced in this manuscript (project title: Targeted Next-Generation Sequencing: a Promising Approach for *Mycobacterium tuberculosis* Detection and Drug Resistance When Applied in Paucibacillary Clinical Samples, accession number PRJNA1238033).

## ETHICS APPROVAL

The study protocol was approved by the Zhongshan Hospital Ethical Review Committee (approval No. B2024-310). Since the study was retrospective and non-interventional, informed consent was not required.

## ADDITIONAL FILES

The following material is available online.

### Supplemental Material

**Fig. S1 (Spectrum03127-24-s0001.tif).** Venn diagram of positive tests for 74 tissue samples.
**Supplemental material (Spectrum03127-24-s0002.docx).** Fig. S1 legend.
**Supplemental tables (Spectrum03127-24-s0003.docx).** Tables S1 to S4.

### Open Peer Review

**PEER REVIEW HISTORY (review-history.pdf).** An accounting of the reviewer comments and feedback.

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
