## [Reviewer comments · Microbiology Spectrum]

Microbiology Spectrum

Targeted next-generation sequencing: a promising approach for *Mycobacterium Tuberculosis* detection and drug resistance when applied in paucibacillary clinical samples

Wenting Jin, Meixia Wang, Yang Wang, Beidi Zhu, Qingqing Wang, Chunmei Zhou, Pei Li, Chaohui Hu, Jun Liu, Jue Pan, Jiachang Chen, and Bijie Hu

Corresponding Author(s): Bijie Hu, Zhongshan Hospital Fudan University

Review Timeline:

Submission Date:	December 13, 2024
Editorial Decision:	January 27, 2025
Revision Received:	February 18, 2025
Editorial Decision:	March 17, 2025
Revision Received:	March 19, 2025
Accepted:	March 31, 2025

Editor: Sadjia Bekal

Reviewer(s): Disclosure of reviewer identity is with reference to reviewer comments included in decision letter(s). The following individuals involved in review of your submission have agreed to reveal their identity: Fangyou Yu (Reviewer #2); Bo Yan (Reviewer #3)

Transaction Report:

DOI: <https://doi.org/10.1128/spectrum.03127-24>

Re: Spectrum03127-24 (Targeted next-generation sequencing: a promising approach for Mycobacterium Tuberculosis detection and drug resistance when applied in paucibacillary clinical samples)

Dear Prof. Bijie Hu:

Thank you for the privilege of reviewing your work. Below you will find my comments, instructions from the Spectrum editorial office, and the reviewer comments.

Revision Guidelines

Sincerely,
Sadjia Bekal
Editor
Microbiology Spectrum

Reviewer #2 (Comments for the Author):

The article has some innovative aspects, and the study includes various sample types, providing certain clinical reference value. However, there are still the following issues that need to be addressed:

1. "Clinical samples" and "clinical specimens" are used interchangeably, as well as "drugs" and "medications."
2. The N values for each item in Figure 1 Routine practice vary greatly; please explain the reasons.

3. In Tables 1 and 2, the sum of different types of specimens is 167, not 178?
4. Do Tables 3 and 4 add up to a total of 200 for respiratory and non-respiratory samples?
5. The first column of Tables S1 and S2 refers to specimens, right? Using PTB and EPTB is inappropriate.
6. Figures 2 and S1A can be deleted.
7. The labels in Figure 3 are unclear and difficult to understand.
8. Does the pie chart in Figure 4A exceed 100%? Please check the data.
9. There is a spelling error in "Xpert" in Figure 5.
10. Genes should be italicized.
11. Replace Xpert with the commercial name.
12. The structure and logic of the discussion are somewhat chaotic and do not deeply explore the results.
13. Where is Table S5 mentioned on line 260?
14. Why "other than Xpert" for MRS on line 126?
15. Line 242: Check the results; should it be 71 positive results?
16. Line 282: Were INH, FQN, SM, ETO, and PTO resistance mutation sites also detected by tNGS?
17. Line 283: *QLN gyrA*?

Reviewer #3 (Comments for the Author):

Since mNGS technique was adopted in clinical settings, the studies about sensitivity and specificity of mycobacterial detection were widely conducted. However, there are still lots of disadvantages when applying mNGS for more patients. tNGS is now a promising way to cut down the high cost of mNGS and perhaps a suitable substitution in the future. Therefore, it is important to test its efficacy in all kinds of pathogenic samples. This article provides useful data from China and supports the adoption of tNGS in TB detection and its drug resistance test, which helps us to evaluate its value in mycobacterial region. Overall, this manuscript is well designed and written. However, there are still several problems that can be solved before being published.

1. It is certain that several detection methods can be combined when applied in clinical settings. Therefore, it would be better if the authors can provide the data of combining tNGS and other methods and calculating its efficacy, giving us more inspirations.
2. Title, the letter T in "Mycobacterium Tuberculosis" should be lowercased.
3. Line 85, a space should be added in "rifampicinresistance".
4. Grammar can be improved.

Responds to the reviews

Thank you for your letter and for providing valuable feedback on our manuscript entitled “Targeted next-generation sequencing: a promising approach for Mycobacterium tuberculosis detection and drug resistance when applied in paucibacillary clinical samples” (ID: Spectrum 03127-24). We deeply appreciate the reviewers’ constructive comments, which have been instrumental in helping us revise and enhance the manuscript.

We have carefully addressed each comment and made the necessary revisions, which we hope will meet your approval.

Reviewer #2 (Comments for the Author):

The article has some innovative aspects, and the study includes various sample types, providing certain clinical reference value. However, there are still the following issues that need to be addressed:

1."Clinical samples" and "clinical specimens" are used interchangeably, as well as "drugs" and "medications."

Respond: We changed "clinical specimens" to "Clinical samples" in Line 44, 93, 109 and "medications" to "drugs" in Line 123 and 124.

2.The N values for each item in Figure 1 Routine practice vary greatly; please explain the reasons 1.

Respond: Since AFB, culture, Xpert, and Pathology tests were standard clinical procedures, several related tests were found to have not been submitted for testing during clinical practice. As a result, no results were obtained. There were 82 tissue specimens, all of which had pathological examination (we found that 82 was

mistakenly written as 83 in Figure 1, which has been verified as 82, consistent with the table S1 and S2).

3. In Tables 1 and 2, the sum of different types of specimens is 167, not 178?

Respond: 178 was the total number of our study, while in the classification of samples, 11 samples of pus were classified into another category, but due to the small amount of size, the sensitivity, specificity, ACC, and AUC results could not be obtained. This was explained in the “Note” in the tables, seen in table 1 and table 2 Note.

4. Do Tables 3 and 4 add up to a total of 200 for respiratory and non-respiratory samples?

Respond: We found that 74 was mistakenly written as 96 in Table 3 and Table 4, which has been verified as 74 seen in table 3 and table 4.

5. The first column of Tables S1 and S2 refers to specimens, right? Using PTB and EPTB is inappropriate.

Respond: We changed “PTB” into “Respiratory” and “EPTB” into “Non respiratory”, seen in table S1 and S2.

6. Figure S2 and S1A can be deleted.

Respond: Figure S2 and S1A was deleted and the corresponding description in the article has also been deleted, Line 247-250. And The relevant subtitle has also been modified, seen in line 246.

7. The labels in Figure 3 are unclear and difficult to understand.

Respond: We have modified Figure 3 with a white background, and we changed “PTB and EPTB” into “Respiratory and Non Respiratory Samples” to help better understand, seen in figure 2.

8. Does the pie chart in Figure 4A exceed 100%? Please check the data.

Respond: We changed the percentage to retain 1 decimal, 44.7% + 36.0% + 19.3% =100%, seen in figure 3A.

9. There is a spelling error in "Xpert" in Figure 5.

Respond: We revised the "Xpet" into "Xpert" in Figure 4.

10. Genes should be italicized.

Respond: The resistant gene mentioned in Methods section was italicized in Line 184-187 ; rpoB was also italicized in Line 273, 346 and 352; katG was also italicized in Line 360. The resistant gene mentioned in Figure 4 were also italicized.

11. Replace Xpert with the commercial name.

Respond: The commercial name was in the Methods section Line 121, referred as Xpert in the following text.

12. The structure and logic of the discussion are somewhat chaotic and do not deeply explore the results.

Respond: We adjusted some sentences and descriptions in the discussion section.

13. Where is Table S5 mentioned on line 260?

Respond: We deleted "table S5" in Line 261 because Table S5 was not included in the supplementary materials.

14. Why "other than Xpert" for MRS on line 126?

Respond: We modified it into a more suitable description "including Xpert and TB-PCR", seen in Line 125.

15. Line 242: Check the results; should it be 71 positive results?

Respond: Figure 2 and S1 was deleted and Line 242- was also deleted.

16.Line 282: Were INH, FQN, SM, ETO, and PTO resistance mutation sites also detected by tNGS?

Respond: We modified it into a more suitable description “INH, FQN, SM, ETO and PTO resistance mutation sites were also detected by tNGS in other samples.”

17.Line 283: QLN gyrA?

Respond: We found it was mistakenly written as “QLN gyrA: Asp94”, which has been verified as “QLN gyrA”.

Reviewer #3 (Comments for the Author):

Since mNGS technique was adopted in clinical settings, the studies about sensitivity and specificity of mycobacterial detection were widely conducted. However, there are still lots of disadvantages when applying mNGS for more patients. tNGS is now a promising way to cut down the high cost of mNGS and perhaps a suitable substitution in the future. Therefore, it is important to test its efficacy in all kinds of pathogenic samples. This article provides useful data from China and supports the adoption of tNGS in TB detection and its drug resistance test, which helps us to evaluate its value in mycobacterial region. Overall, this manuscript is well designed and written. However, there are still several problems that can be solved before being published.

1.It is certain that several detection methods can be combined when applied in clinical settings. Therefore, it would be better if the authors can provide the data of combining tNGS and other methods and calculating its efficacy, giving us more inspirations.

Respond: Table S3 and Table S4 showed “the diagnosis performance of combination of Xpert and tNGS for tuberculosis in diverse sample types in comparison to CRS and MRS standard” respectively.

2.Title, the letter T in "Mycobacterium Tuberculosis" should be lowercased.

Respond: The initial letter T of “Tuberculosis” is changed to lowercase in line 2. We also modified “Mycobacterium Tuberculosis (MTB)” to “Mycobacterium tuberculosis (MTB)” in line 37 and “Mycobacterium tuberculosis” was also italicized in line 165.

3. Line 85, a space should be added in "rifampicinresistance".

Respond: A space was added between rifampicin and resistance.

4. Grammar can be improved.

Respond: We adjusted some sentences and descriptions in the discussion section.

Re: Spectrum03127-24R1 (Targeted next-generation sequencing: a promising approach for Mycobacterium Tuberculosis detection and drug resistance when applied in paucibacillary clinical samples)

Dear Prof. Bijie Hu:

Thank you for the privilege of reviewing your work. Below you will find my comments, instructions from the Spectrum editorial office

If any sequencing data is generated, it should be deposited at NCIB and the accession number (Project number), should be added into the text

Sincerely,
Sadjia Bekal
Editor
Microbiology Spectrum

Responds to the reviews

Thank you for your letter and for providing valuable feedback on our manuscript entitled “Targeted next-generation sequencing: a promising approach for Mycobacterium tuberculosis detection and drug resistance when applied in paucibacillary clinical samples” (ID: Spectrum 03127-24). We deeply appreciate the reviewers’ constructive comments, which have been instrumental in helping us revise and enhance the manuscript.

We have carefully addressed each comment and made the necessary revisions, which we hope will meet your approval.

Reviewer #2 (Comments for the Author):

The article has some innovative aspects, and the study includes various sample types, providing certain clinical reference value. However, there are still the following issues that need to be addressed:

1. "Clinical samples" and "clinical specimens" are used interchangeably, as well as "drugs" and "medications."

Respond: We changed "clinical specimens" to "Clinical samples" in Line 44, 93, 109 and "medications" to "drugs" in Line 123 and 124.

2. The N values for each item in Figure 1 Routine practice vary greatly; please explain the reasons 1.

Respond: Since AFB, culture, Xpert, and Pathology tests were standard clinical procedures, several related tests were found to have not been submitted for testing during clinical practice. As a result, no results were obtained. There were 82 tissue specimens, all of which had pathological examination (we found that 82 was

mistakenly written as 83 in Figure 1, which has been verified as 82, consistent with the table S1 and S2).

3. In Tables 1 and 2, the sum of different types of specimens is 167, not 178?

Respond: 178 was the total number of our study, while in the classification of samples, 11 samples of pus were classified into another category, but due to the small amount of size, the sensitivity, specificity, ACC, and AUC results could not be obtained. This was explained in the “Note” in the tables, seen in table 1 and table 2 Note.

4. Do Tables 3 and 4 add up to a total of 200 for respiratory and non-respiratory samples?

Respond: We found that 74 was mistakenly written as 96 in Table 3 and Table 4, which has been verified as 74 seen in table 3 and table 4.

5. The first column of Tables S1 and S2 refers to specimens, right? Using PTB and EPTB is inappropriate.

Respond: We changed “PTB” into “Respiratory” and “EPTB” into “Non respiratory”, seen in table S1 and S2.

6. Figure S2 and S1A can be deleted.

Respond: Figure S2 and S1A was deleted and the corresponding description in the article has also been deleted, Line 247-250. And The relevant subtitle has also been modified, seen in line 246.

7. The labels in Figure 3 are unclear and difficult to understand.

Respond: We have modified Figure 3 with a white background, and we changed “PTB and EPTB” into “Respiratory and Non Respiratory Samples” to help better understand, seen in figure 2.

8. Does the pie chart in Figure 4A exceed 100%? Please check the data.

Respond: We changed the percentage to retain 1 decimal, 44.7% + 36.0%+ 19.3% =100%, seen in figure 3A.

9. There is a spelling error in "Xpert" in Figure 5.

Respond: We revised the "Xpet" into "Xpert" in Figure 4.

10. Genes should be italicized.

Respond: The resistant gene mentioned in Methods section was italicized in Line 184-187 ; rpoB was also italicized in Line 273, 346 and 352; katG was also italicized in Line 360. The resistant gene mentioned in Figure 4 were also italicized.

11. Replace Xpert with the commercial name.

Respond: The commercial name was in the Methods section Line 121, referred as Xpert in the following text.

12. The structure and logic of the discussion are somewhat chaotic and do not deeply explore the results.

Respond: We adjusted some sentences and descriptions in the discussion section.

13. Where is Table S5 mentioned on line 260?

Respond: We deleted "table S5" in Line 261 because Table S5 was not included in the supplementary materials.

14. Why "other than Xpert" for MRS on line 126?

Respond: We modified it into a more suitable description "including Xpert and TB-PCR", seen in Line 125.

15. Line 242: Check the results; should it be 71 positive results?

Respond: Figure 2 and S1 was deleted and Line 242- was also deleted.

16.Line 282: Were INH, FQN, SM, ETO, and PTO resistance mutation sites also detected by tNGS?

Respond: We modified it into a more suitable description “INH, FQN, SM, ETO and PTO resistance mutation sites were also detected by tNGS in other samples.”

17.Line 283: QLN gyrA?

Respond: We found it was mistakenly written as “QLN gyrA: Asp94”, which has been verified as “QLN gyrA”.

Reviewer #3 (Comments for the Author):

Since mNGS technique was adopted in clinical settings, the studies about sensitivity and specificity of mycobacterial detection were widely conducted. However, there are still lots of disadvantages when applying mNGS for more patients. tNGS is now a promising way to cut down the high cost of mNGS and perhaps a suitable substitution in the future. Therefore, it is important to test its efficacy in all kinds of pathogenic samples. This article provides useful data from China and supports the adoption of tNGS in TB detection and its drug resistance test, which helps us to evaluate its value in mycobacterial region. Overall, this manuscript is well designed and written. However, there are still several problems that can be solved before being published.

1.It is certain that several detection methods can be combined when applied in clinical settings. Therefore, it would be better if the authors can provide the data of combining tNGS and other methods and calculating its efficacy, giving us more inspirations.

Respond: Table S3 and Table S4 showed “the diagnosis performance of combination of Xpert and tNGS for tuberculosis in diverse sample types in comparison to CRS and MRS standard” respectively.

2.Title, the letter T in "Mycobacterium Tuberculosis" should be lowercased.

Respond: The initial letter T of “Tuberculosis” is changed to lowercase in line 2. We also modified “Mycobacterium Tuberculosis (MTB)” to “Mycobacterium tuberculosis (MTB)” in line 37 and “Mycobacterium tuberculosis” was also italicized in line 165.

3. Line 85, a space should be added in "rifampicinresistance".

Respond: A space was added between rifampicin and resistance.

4. Grammar can be improved.

Respond: We adjusted some sentences and descriptions in the discussion section.

Re: Spectrum03127-24R2 (Targeted next-generation sequencing: a promising approach for Mycobacterium Tuberculosis detection and drug resistance when applied in paucibacillary clinical samples)

Dear Prof. Bijie Hu:

Your manuscript has been accepted, and I am forwarding it to the ASM production staff for publication. Your paper will first be checked to make sure all elements meet the technical requirements. ASM staff will contact you if anything needs to be revised before copyediting and production can begin. Otherwise, you will be notified when your proofs are ready to be viewed.

Sincerely,
Sadjia Bekal
Editor
Microbiology Spectrum

Reviewer #2 (Comments for the Author):

It has been revised according to the modification suggestions. However, attention should still be paid to the details. For example, there is the mixed use of Chinese and English punctuation marks, and some grammatical errors need to be corrected.

Reviewer #3 (Comments for the Author):

All my concerns have been addressed, and the MS is now ready for publication.

**Targeted next-generation sequencing: a promising approach**
**for *Mycobacterium tuberculosis* detection and drug**
**resistance when applied in paucibacillary clinical samples**

Wenting Jin^{1, #}, Meixia Wang^{2, #}, Yang Wang^{3, #}, Beidi Zhu¹, Qingqing Wang¹,
Chunmei Zhou⁴, Pei Li⁵, Chaohui Hu³, Jun Liu³, Jue Pan¹, Jiachang Chen^{3,*}, Bijie Hu^{1,*}

¹ Department of Infectious Diseases, Zhongshan Hospital, Fudan University, Shanghai,
China

² Department of Infection management, Zhongshan Hospital (Xiamen), Fudan
University, Xiamen, Fujian, China

³ Guangzhou KingCreate Biotechnology Co., Ltd., Guangzhou, China

⁴ Department of Microbiology, Zhongshan Hospital, Fudan University, Shanghai
200032, China

⁵ Kingmed diagnostic, Guangzhou, China

[#] Contributed equally to this work (co-first authors).

17 ^{*} Contributed equally to this work (co-corresponding authors).

Number of words in the abstract: 225

Number of words in the importance: 109

Total word count for the body of the manuscript: 3,461

[revised manuscript text omitted]

(Cepheid, USA), liquid culture (MGIT 960, BD, USA), AFB and histopathology were
all standard diagnostic processes. Clinical samples were stored at -80°C and were
tested for mNGS and MTB-tNGS (referred to as tNGS). Microbiological reference
standard (MRS) referred to positive culture or positive nucleic acid test for MTBC
(including Xpert and TB-PCR). Composite reference standard (CRS) referred to a
patient who presents with symptoms, signs, images, microbial results or pathology
suggestive of TB, where a clinician has diagnosed TB and decided to treat the patient
with a full course of TB therapy. The CRS standard complied with the People's
Republic of China Health Industry Standard (WS 288-2017).

**Xpert MTB/RIF assay**

An equal volume of sample processing reagent was combined with the
homogenate of lymph node tissue, vortexed for 15 seconds, and then left to stand at
room temperature for 15 minutes. The GeneXpert Infinity System (Cepheid, USA)
was then loaded with a sample of two milliliters of the processed liquid that was
added to the Xpert reaction reagent cartridge (Xpert, Cepheid, USA). The findings
were automatically read out by the system after two hours.

**MTB Liquid Culture**

The samples were to be digested, decontaminated, and concentrated using the
CLSI standard methodology. After processing, 0.5 ml of the material is utilized to
quickly cultivate mycobacteria (MGIT 960, BD, USA). On a positive culture sample,
MPB64 antigen detection and acid-fast staining confirmation were carried out.

**mNGS**

The mNGS process was performed on a rapid on-site platform in the hospital, as
described previously (18). In short, clinical samples(0.5mL) were treated to produce
DNA fragments prior to the extraction of genomic DNA. After that, DNA libraries
were created using the BGISEQ-2000 platform, and a bioinformatics pipeline was
used to examine the sequencing results. The RefSeq database was used to align the
resulting data. According to the preceding description, the RefSeq was deemed
positive if at least one read was mapped to the MTBC (number of sequences
rigorously aligned at the genus level, SMRNG ≥ 1).

**MTB-tNGS (tNGS)**

Prior to nucleic acid extraction, sputum samples and viscous BALF were treated
to achieve liquefaction. Fresh tissue samples were minced and homogenized by
oscillation. Processed samples were aliquoted (1.3mL) and subjected to high-speed
centrifugation. The supernatant was then removed, retaining approximately 500 μ L of
the sample. A 490 μ L portion was combined with 10 μ L of an exogenous internal
control and processed in a tissue homogenizer for mechanical disruption. The mixture
was subsequently centrifuged at 12,000rpm for 5minutes, and 250 μ L of the
supernatant was collected for nucleic acid extraction using appropriate extraction or
purification reagents (KingCreate Co. Ltd., Guangzhou, China).

Library construction was performed using a *Mycobacterium tuberculosis*
complex and drug-resistance gene Extraction Kit (KingCreate Co. Ltd., Guangzhou,

China). The extracted nucleic acids were enriched for target regions and underwent
library purification steps to complete the library construction process. Nuclease-free
water was used as a non-template control (NTC) to monitor for contamination.

Generated libraries were quantified using Equalbit DNA HS Assay Kit (Vazyme
Biotech, Nanjing, Jiangsu, China) with Invitrogen™ Qubit™ 3.0/4.0 (Thermo Fisher
Scientific, Waltham, MA, USA) Fluorometer to ensure all samples were with library
density $\geq 0.5\text{ng}/\mu\text{L}$ or else the library should be subjected to re-construction. The
constructed libraries were pooled to homogeneous mass. The size of the library
fragments was determined by an automated nucleic acid protein analyzer (Qsep100)
using Standard Cartridge Kit (S2). The size of the library fragments should be from
250 to 350bp. Qualified pooled library was diluted and denatured, 500 μL of which
was subjected to KM MiniSeq Dx-CN Platform (KingCreate Co. Ltd., Guangzhou,
China) for sequencing.

Generated sequencing raw read data were undergone quality control procedure.
The fastp v0.20.1 (3) was employed for adaptor trimming (19) and low-quality
filtering using default parameters followed by reference-based assembly using bwa
v0.7.17 in 'mem' mode(20). The tNGS assay interrogates 18 genes using
631amplicons from loci associated with MTBC and resistance to 13 anti-TB drugs:
RIF (*rpoB*), INH (*inhA* and *katG*), PZA (*pncA*), EMB (*embB* and *embA*),
fluoroquinolones (*gyrA* and *gyrB*), LZD (*rplC* and *rpl*), BDQ (*atpE* and *Rv0678*), CFZ
(*Rv0678*), SM (*rrs*, *rpsL* and *qid*), KAN and AMK (*rrs* and *eis*), CPM (*rrs*, *rplA*),
ETO (*ethA* and *inhA*) and PTO (*inhA*), mainly from Genbank and Refseq. To call
positive signals for specific pathogens, mapped reads were counted and normalized to
read per 100,000 reads (RPhK). Cases with specific RPhKs were considered as
positive for each sample. If a specific species or high-level taxonomy unit was
identified in a sample with RPhK value ≥ 3 , this species / unit was regarded as
“present” in this sample, or else reported as “absent”.

**Sanger Sequencing**

The extracted nucleic acid was added to the Sanger reaction mixture for PCR
amplification. After amplification, the product was purified and electrophoresed to
confirm the target band, which was then excised and used for Sanger sequencing to
obtain sequence data.

**Statistical analysis**

Continuous variables were described as mean±standard deviation (SD) and
median and interquartile range (IQR) according to data distribution. The
Mann-Whitney U test or t test was used to compare the characteristics between two
groups. The diagnostic effectiveness was assessed by computing the sensitivity,
specificity, accuracy (ACC) and AUC value for various methods. Kappa analysis was
used to describe the level of data consistency. $P < 0.05$ with two-tailed was
considered to be statistically significant. Statistical analyses were performed using the
R-3.6.1 software.

**RESULT**

**Characteristics of participants and samples**

Totally, 178 participants (105 male/73 female, average age of 55.1 ± 17.84 , the
male to female ratio of 1.44:1) were included in this study. The flowchart of the study
was shown in Figure 1. Among all the samples, 10 were AFB positive. Due to the
diverse sources of samples, we classify them by three ways: respiratory/non
respiratory, tissue/non tissue, as well as more detailed sputum, BALF, lung tissue,
extrapulmonary tissue, serosal fluid and pus. The specific distribution is shown in
Table 1-2, Table S1-S2.

**Performance of different methods in detecting MTB**

The overall sensitivity of culture, Xpert, mNGS, and tNGS were 0.458 [95%
confidence interval (95% CI): 0.377-0.540], 0.614 (0.534-0.695), 0.772 (0.704-0.839)
and 0.760 (0.692-0.828), respectively in comparison to CRS. They all had specificity
of 1. tNGS had similar sensitivity as mNGS, which had advantages over culture and
Xpert receptively despite of different classification of sample types. Outstandingly,
the advantage was more pronounced and also surpassed mNGS in serous fluid, as
shown in Table 1 and Table S1. The overall sensitivity of culture, Xpert, mNGS, and
tNGS were 0.606 (95% CI: 0.514-0.697), 0.811 (0.737-0.886), 0.856 (0.791-0.921)
and 0.884 (0.825-0.943), respectively, in comparison to MRS, as shown in Table 2
and Table S2. Culture and Xpert both had specificity of 1, while mNGS and tNGS
had a specificity of 0.697 (0.586-0.808) and 0.773 (0.672-0.874). In sputum, BALF
and pulmonary tissue sample, the sensitivity of Xpert was not significantly lower than

[revised manuscript text omitted]

Reference

- 1. WHO. 2024. Global tuberculosis report 2024.
2. Zhang L, Weng TP, Wang HY, Sun F, Liu YY, Lin K, Zhou Z, Chen YY, Li
YG, Chen JW, Han LJ, Liu HM, Huang FL, Cai C, Yu HY, Tang W, Huang
ZH, Wang LZ, Bao L, Ren PF, Deng GF, Lv JN, Pu YL, Xia F, Li T, Deng Q,
He GQ, Li Y, Zhang WH. 2021. Patient pathway analysis of tuberculosis

- diagnostic delay: a multicentre retrospective cohort study in China. *Clin*
*Microbiol Infect* 27:1000-1006.
- 3. Schön T, Matuschek E, Mohamed S, Utukuri M, Heysell S, Alffenaar J-W,
Shin S, Martinez E, Sintchenko V, Maurer FP, Keller PM, Kahlmeter G,
Köser CU. 2019. Standards for MIC Testing That Apply to the Majority of
Bacterial Pathogens Should Also Be Enforced for Mycobacterium
Tuberculosis Complex. *Clinical microbiology and infection : the official*
*publication of the European Society of Clinical Microbiology and Infectious*
*Diseases* 25:403-405.
- 4. WHO. 2014. Xpert MTB/RIF implementation manual: technical and
operational ‘how-to’; practical considerations.
- 5. Miotto P, Tessema B, Tagliani E, Chindelevitch L, Starks AM, Emerson C,
Hanna D, Kim PS, Liwski R, Zignol M, Gilpin C, Niemann S, Denkinger CM,
Fleming J, Warren RM, Crook D, Posey J, Gagneux S, Hoffner S, Rodrigues
C, Comas I, Engelthaler DM, Murray M, Alland D, Rigouts L, Lange C,
Dheda K, Hasan R, Ranganathan UDK, McNerney R, Ezewudo M, Cirillo
DM, Schito M, Köser CU, Rodwell TC. 2017. A standardised method for
interpreting the association between mutations and phenotypic drug resistance
in Mycobacterium tuberculosis. *European Respiratory Journal* 50.
- 6. Allix-Beguec C, Arandjelovic I, Bi L, Beckert P, Bonnet M, Bradley P,
Cabibbe AM, Cancino-Munoz I, Caulfield MJ, Chaiprasert A, Cirillo DM,
Clifton D, Comas I, Crook DW, De Filippo MR, de Neeling H, Diel R,
Drobniewski FA, Faksri K, Farhat MR, Fleming J, Fowler P, Fowler TA, Gao
Q, Gardy J, Gascoyne-Binzi D, Gibertoni-Cruz A-L, Gil-Brusola A,
Golubchik T, Gonzalo X, Grandjean L, He G, Guthrie JL, Hoosdally S, Hunt
418 M, Iqbal Z, Ismail N, Johnston J, Khanzada FM, Khor CC, Kohl TA, Kong C,
Lipworth S, Liu Q, Maphalala G, Martinez E, Mathys V, Merker M, Miotto P,

[revised manuscript text omitted]

Tuberculosis and respiratory diseases 78:47-55.
- 42. Kang WL, Yu JJ, Du J, Yang S, Chen HY, Liu JX, Ma JS, Li MW, Qin JM,
Shu W, Zong PL, Zhang Y, Dong YK, Yang ZY, Mei ZX, Deng QY, Wang P,
Han WG, Wu MY, Chen L, Zhao XG, Tan L, Li FJ, Zheng C, Liu HW, Li XJ,
Ertai A, Du YR, Liu FL, Cui WY, Wang QH, Chen XH, Han JF, Xie QY,
Feng YM, Liu WY, Tang PJ, Zhang JY, Zheng J, Chen DW, Yao XY, Ren T,
Li Y, Li YY, Wu L, Song Q, Yang M, Zhang J, Liu YY, Guo SL, et al. 2020.

The epidemiology of extrapulmonary tuberculosis in China: A large-scale
multi-center observational study. Plos One 15.

**Figure legends**

Figure 1: Flowchart of study design.

Figure 2: Scatter plot of TB sequencing (log 10) detected by tNGS and mNGS based
on different Xpert semi-quantitative categories.

Xpert: Xpert MTB/RIF; PTB: pulmonary tuberculosis; EPTB: extrapulmonary
tuberculosis; tNGS: targeted next-generation sequencing; mNGS: metagenomic
next-generation sequencing; RPhK: mapped reads per 100,000; SMRNG: number of
sequences rigorously aligned at the genus level.

Figure 3: Drug resistant related mutations detected in 114 tuberculosis samples.

Figure 4: Drug resistant related mutations in 15 cases using Xpert MTB/RIF, tNGS
and sanger sequencing.

RIF: rifampicin; INH: isoniazid; EMB: ethambutol; PZA: pyrazinamide; FQN:
fluoroquinolone; ETO; ethionamide; PTO: prothionamide; RPhK: mapped reads per
100,000
